# The Efficacy and Safety of the GATT Procedure in Open-Angle Glaucoma—6-Month Results

**DOI:** 10.3390/ijerph20032759

**Published:** 2023-02-03

**Authors:** Agnieszka Ćwiklińska-Haszcz, Tomasz Żarnowski, Dominika Wróbel-Dudzińska, Ewa Kosior-Jarecka

**Affiliations:** Department of Diagnostics and Microsurgery of Glaucoma, Medical University of Lublin, 20-059 Lublin, Poland

**Keywords:** glaucoma surgery, angle surgery, gonioscopy-assisted transluminal trabeculotomy, GATT, minimally invasive glaucoma surgery, MIGS, trabeculotomy ab interno

## Abstract

AIM. The aim of the study was to retrospectively evaluate the efficacy and safety of GATT during a 6-month observation period. MATERIAL AND METHODS. The studied group consisted of 69 open-angle glaucoma patients treated with GATT as the only procedure or in combination with cataract surgery. Patients were assessed 1 day, 10 days, 1 month, 3 months, and 6 months after the surgery via standard ophthalmic examination including VF, IOP, and BCVA. The number of medications taken daily and possible complications were checked. Two criteria of success were established (S1: IOP decrease by 30% and S2: IOP lower than 18 mm Hg). RESULTS. Before the surgery, the mean IOP was 26.94 mmHg and significantly decreased after GATT to 15.59 mmHg at 6M. BCVA did not significantly differ between the visits. The mean MD did not change significantly within the 6-month observation period (−8.20 dB vs. −8.16 dB, *p* = 0.9824), similar to the mean VFI (64.31% vs. 63.05%, *p* = 0.8571). A 30% IOP decrease at 6M visit was obtained in 95.6% of patients, and 37.7% needed medications to stabilize IOP. An IOP lower than 18 mmHg at 6M was obtained in 91.3% of studied patients after the GATT procedure, and in 58.0% without additional medications. The mean number of medications received daily decreased significantly at 6M compared to preoperative results (2.59 at inclusion vs. 0.76 at 6M, *p* = 0.0004). The most frequent complication after surgery was hyphema, which resolved spontaneously within 10 days. CONCLUSION. The 6-month observation showed that GATT is a minimally invasive glaucoma-surgery technique that enables an effective and safe IOP decrease.

## 1. Introduction

All procedures effective in slowing the progression of glaucoma are based on the reduction of intraocular pressure (IOP), which can be obtained by medical, laser, or surgical treatment [1]. For more than 50 years, trabeculectomy [2] has remained the gold standard in the surgical management of glaucoma. It is a procedure in which a new outflow pathway is created for aqueous humor to the subconjunctival space. However, despite its great efficacy in long-term high IOP lowering, it can also cause a lot of complications, mainly related to the created bleb [3,4]. Additionally, in the early stages of glaucoma or in older patients, low-teen IOP is not always needed [5]. This is why new techniques are being developed to improve the safety profile of glaucoma surgery and induce a reasonable decrease in IOP.

Blebless methods focus on the improvement of aqueous-humor outflow through the natural pathway, starting with the Schlemm’s canal. During the surgery, the Schlemm’s canal can be reached from a scleral approach (ab externo) or from the anterior chamber (ab interno). Ab externo procedures need a conjunctival and scleral opening, and after reaching the Schlemm’s canal lumen it can be dilatated by the injection of viscoelastic material and suture tension (as in canaloplasty), or its internal wall may be removed (360° trabeculotomy). The ab interno approach spares conjunctiva and sclera and the techniques are referred to as minimally invasive glaucoma surgery (MIGS) methods. There is also a set of ab interno methods with canal-lumen viscodilatation (canaloplasty), implant injection (iSTENT or Focus), local incision of the internal wall (goniotomy with Trabex or Kahook Dual Blade knife), or extended removal of the internal wall, and they include gonioscopy-assisted transluminal trabeculotomy (GATT).

The GATT technique introduced by Grover et al. [6] consists of peeling the interior wall of the Schlemm’s canal and perfectly matches the pathogenesis of IOP increase in glaucoma, which is related to the enhanced outflow resistance at this site. Since its introduction, the application of the GATT technique has been described in different types of glaucoma [7,8,9,10]. 

The aim of this study is to evaluate the efficacy and safety of GATT within a short observation period.

## 2. Material and Methods

### Studied Group

This study presents a retrospective analysis of the results of GATT (alone or combined with cataract surgery) in 69 patients with open-angle glaucoma performed in the years 2021–2022 at the Department of Diagnostics and Microsurgery of Glaucoma, Medical University of Lublin, Poland. All patients with primary open-angle glaucoma or pseudoexfoliative glaucoma who underwent GATT and had at least 6 months of observation were included. The exclusion criteria were as follows: neovascular glaucoma, traumatic glaucoma, narrow or closed angle in gonioscopy, normal-tension glaucoma, secondary glaucoma different than PEXG, age under 18 years. The combined procedure was performed in 41 cases (59.4%), and in 28 (40.6%) cases GATT was performed as a standalone procedure. Eight patients had undergone previous antiglaucoma surgery; 4 of them had more than one procedure. Demographic characteristics of the studied group are presented in Table 1.

The patients were examined 1 day, 7 days, 1 month, 3 months, and 6 months after the procedure. At each visit, BCVA (Snellen decimal scale), IOP (Goldman applanation tonometry), slit-lamp examination, and binocular ophthalmoscopy were performed, and surgery success was assessed according to the established criteria. The number of medications taken was checked. Antiglaucoma drops were introduced when the success criteria were not fulfilled. 

Two criteria were established to assess the success of the surgery:

S1: a 30% decrease in IOP as compared to the value measured at the inclusion visit;

S2: IOP ≤ 18 mm Hg.

Both criteria were reported as a total success (“S1” or “S2”) when they were met without medication, or as a qualified success (“Q1” or “Q2”) when they were fulfilled with additional medical treatment. Additionally, the procedure was considered a failure when additional surgical treatment was needed to obtain a successful IOP level.

## 3. Methods

### 3.1. GATT Procedure

A clear corneal incision of 1.6 mm was performed at the temporal side of the eye. The anterior chamber was filled with viscoelastic material and goniotomy was performed at the opposite side of the angle, spanning approximately two clock hours of the angle. The Schlemm’s canal was then cannulated with a blunted Prolene 6/0 suture along 360 degrees. If cannulation in one direction did not successfully encircle the whole of the Schlemm’s canal’s circumference, cannulation in the other direction was attempted from the same goniotomy. The inner wall of the Schlemm’s canal was peeled off, causing unroofing of the canal. This was followed by anterior-chamber irrigation and injection of intracameral antibiotics. 

### 3.2. Combined Procedure—GATT with Cataract Extraction

A standard phacoemulsification procedure was performed at the temporal side of the eye with a 2.2 mm clear corneal incision. Posterior-chamber intraocular-lens implantation was followed by miochol injection. The GATT procedure was performed as previously described.

Cataract surgery was performed by one of two experienced surgeons (AĆH or TŻ). All GATT procedures were performed by EKJ. The presented results are a retrospective analysis initiated when the surgeons’ technique was stabilized. 

## 4. Results

Before the surgery, the mean IOP was 26.94 mmHg and was significantly decreased by the GATT procedure to a mean value of 15.59 mmHg. Statistical significance was observed when comparing all postoperative values to preoperative values; no significance was found among postoperative values. The details of IOP changes during the study are presented in Figure 1. The details of changes in BCVA during the study are shown in Figure 2. An (insignificant) decrease in BCVA was observed in 1D postoperatively. The mean MD did not change significantly within 6 months of observation (−8.20 dB vs. −8.16 dB, *p* = 0.9824), similar to the mean VFI (64.31% vs. 63.05%, *p* = 0.8571).

As far as the criteria of success are concerned, a 30% IOP decrease after one month was obtained in 95.6% of patients. However, 37.7% of patients needed medications to stabilize IOP. Failure at 6M was observed in three patients (4.4%), who needed an additional surgical procedure to decrease IOP. The details of changes in the number of patients fulfilling criterion S1 are presented in Table 2.

An IOP lower than 18 mm Hg at 6M was obtained in 91.3% of patients after the GATT procedure, including in 58.0% without additional medications. The details are shown in Table 3.

The mean number of medications taken daily decreased significantly at 6M compared to preoperative results (2.59 drops at inclusion vs. 0.76 drops at 6M, *p* = 0.0004).

In two cases, the surgical procedure was complicated by iridodialysis caused by accidentally disrupting the iris with forceps; one patient needed a surgical intervention on day 1 after the surgery. A common early complication observed in about 50% of patients was hyphema during the first few days after the surgery. The blood in the anterior chamber influenced the results of early post-op BCVA. At the visit 10 days later, hyphema was found to have resolved spontaneously in every case without additional interventions, and BCVA had significantly increased. In three patients with advanced glaucoma, we observed transient visual loss in which the BCVA decreased to hand movement in front of the eyes. All of these patients were carefully monitored during the early postoperative hours and IOP spikes were excluded. Hyphema was visible in AC on day 1. However, 10 days later, BCVA had not improved despite its resolution, although IOP remained in the low teens. BCVA returned to the preoperative values at the 1-month visit in one patient and at the 3-month visit in two patients. Corneal oedema was reported in two cases: in a POAG patient, it was observed on day 1 after surgery and it resolved before the 10-day visit; and in the other case corneal oedema was observed in a PEXG patient at the 10-day visit and spontaneously resolved by the 1-month visit. During the short observation period, three patients needed additional antiglaucoma surgery. In all cases, trabeculectomy was successfully performed 3 months after the GATT procedure. A summary of the complications is shown in Table 4.

## 5. Discussion

The GATT procedure was designed to be performed as a standalone surgery or in combination with cataract extraction. When introducing the technique, Grover et al. [6] proposed that the GATT surgery be performed first before the cataract removal. In this study, this order was changed. The primary cataract surgery enables the surgeon to deepen the anterior chamber and facilitate access to the iridocorneal angle in goniotomy. Additionally, during the hypotony that accompanies the opening of the eye, blood reflux enables proper localization of the Schlemm’s canal. If performed first, successful goniotomy usually causes bleeding to the anterior chamber, which may disturb visualization during cataract surgery. On the other hand, the presence of the lens stabilizes the anterior-chamber volume during the GATT surgery, and as a result, less viscoelastic material can be used.

The best option for patients who need glaucoma surgery and have concomitant surgical cataracts is a combination of the procedures [11,12,13]. When opting for a single procedure, the need for the additional surgery within a short period and the sequence of surgeries is discussed, as both options have their disadvantages. One of the procedures offered to this group of patients is phacotrabeculectomy [14,15]. However, clinical studies have shown inferiority in the efficacy of trabeculectomy combined with cataract extraction [16,17]. Still, cataract surgery alone is thought to be one of the MIGS procedures with IOP-lowering potential even in open-angle glaucoma. The results of GATT, whether performed alone or in combination with other procedures, did not differ significantly [18], which shows that opening of the inner wall of the Schlemm’s canal has an IOP-lowering potential independently of deepening the anterior chamber caused by lens exchange.

The mean decrease in IOP observed in our study was 11.4 mmHg during a 6-month analysis, results similar to those reported by other authors [19,20,21]. The decline reported in a meta-analysis [22] amounted to 9.81 mmHg and was independent of the method of canalization of the Schlemm’s canal (catheter or different types of sutures). 

A study by Fontana et al. [23] compared the 18-month efficacy of GATT and trabeculectomy in patients with uncontrolled open-angle glaucoma and showed that the success rate according to the same criteria as those adopted in our study was statistically similar in both groups (complete and qualified success rates were 59% and 27% after TRAB, and 46% and 31% after GATT, respectively). However, the mean IOP obtained in both procedures (15.26 ± 3.47 mmHg after GATT and 12.48 ± 4.58 mmHg after trabeculectomy), as well as the percentage decrease from baseline in IOP (56.05 ± 17.72 after trabeculectomy and 42.04 ± 15.56 after GATT), were in favor of trabeculectomy. In GATT, the removal of the inner layer of the Schlemm’s canal, where the majority of the pathologic outflow resistance takes place, makes it possible to lower IOP to episcleral venous pressure, known to be approximately 12 mmHg [24]. A pathological increase in outflow resistance in collector channels during glaucoma and the scarring of the trabeculotomy opening result in a final IOP in the range of mid-teen values. In advanced glaucoma and some cases of NTG, the target IOP is in the low teens, which probably cannot be obtained with GATT in long-term observation.

Most glaucoma patients usually receival medical therapy in the form of daily multidose treatment. Eyedrops frequently have potentially harmful side effects, such as low-grade inflammation, dry-eye syndrome, and dermatoconjunctivitis [25]. Additionally, their prolonged application diminishes the surgical success of planned antiglaucoma surgeries involving the conjunctiva. In our study, the GATT procedure enabled a significant decrease in the number of drops taken daily and about 60% of patients did not need any additional medical therapy.

One of the major advantages of the GATT procedure is its safety. The most frequently described complications were hyphema, transient hypotony, IOP spikes, and corneal oedema. In our group of patients, we did not observe any sight-threatening complications. Additionally, no cases of hypotony were observed. We did not intentionally leave viscoelastic material in the anterior chamber after GATT, instead trying to remove all the remnants at the end of the procedure. This may be the reason why early postoperative IOP spikes were not observed in our group. On the other hand, it may have increased the incidence of early hyphemas.

The most frequently described complication was hyphema the day following the procedure. In all cases it resolved spontaneously within a week. The incidence of reported hyphemas varies from 12.5% [26] up to 80% of cases [27], with 50% observed in our group. Previous studies on this technique described hyphema as an inevitable event that occurs as a result of tearing the vascularized-angle structures [28]. However, in Schlemm’s-canal surgery, hyphema is described as a good predictive sign of surgical success since it shows the functioning of the connection between the anterior chamber and the distal-outflow pathways for aqueous humor. In our patients, we tried to avoid leaving viscoelastic material in the AC at the end of the surgery. Theoretically, it may limit the bleeding, but on the other hand, it may result in IOP peaks in the early postoperative period, which can have a deleterious effect on RNFL, especially in advanced glaucoma. To minimize the amount of blood in the AC, we tried to avoid hypotony at the end of the surgery and leave the bulb a little hypertonus. Hyphema typically decreases visual acuity the day following the surgery, so patients should be informed about it. 

Among our patients, we had some cases of very advanced glaucoma. In spite of uncomplicated surgery and strict IOP monitoring during the early postoperative period, which did not show any peaks, we observed transient visual loss in three patients, which lasted up to 1 month after surgery. This complication of GATT has never been reported before. At first, it was diagnosed as a wipe-out syndrome. However, improvement in visual acuity in this phenomenon has not been described, whereas in our group, BCVA improved in the end.

Corneal edema, also described by Grover et al. [18], was observed in two patients in the early postoperative period, and it resolved spontaneously. However, the fact that manipulation in the AC during GATT may cause a decrease in the number of endothelial cells, especially when combined with cataract surgery, requires further studies. The 1-month results showed that GATT appears to be a safe procedure for the corneal endothelial-cell layer when performed either isolated or combined with cataract extraction [29].

Less frequent complications described after GATT were Descemet’s detachment, iritis, iridodialysis, and goniosynechiae. In our case series, the surgery was complicated by iridodialysis in two patients, one of whom needed surgical suturing, which did not influence the final IOP-lowering success. Additionally, one highly myopic patient had transient vitreous hemorrhage, probably caused by an incorrectly placed suture.

According to our study, the GATT procedure may be successfully when applied in a variety of open-angle glaucoma patients. Some of our patients had previously had at least one antiglaucoma surgery performed. In the case of unsuccessful IOP decrease in this group, subsequently performed conjunctiva-based procedures were related to the limited success rate. Our study shows that GATT can be successfully applied to ultimately lower IOP or to prolong the time until the next surgery. In the latter case, it is significant for minimizing conjunctival inflammation. Despite doubts related to studies describing the decrease in size of the Schlemm’s canal after successful filtration surgery [30], the clinical data confirm the success of angle-based procedures in patients after previous incisional glaucoma surgeries [28,31,32]. Grover reported a success rate of 60–70%, mainly in the decrease of IOP and not in the number of drops taken. To achieve the latter, it is necessary to obtain low-target IOP in advanced glaucoma cases, which cannot be accomplished by angle surgery alone. This researcher notes that in most cases of previous surgery, an extended opening of the canal was possible, which is similar to our observation. Cubuk et al. [32] found that the results of GATT as the second procedure were better in PEXG than in POAG patients, which is similar to the results in naive patients [7].

When choosing the type of primary antiglaucoma procedure, a lot of different factors need to be considered. Angle procedures seem to have a good safety profile; however, the mean IOP that can be obtained is about 15 mmHg, which should be taken into consideration when planning surgery in patients with advanced neuropathy or normal-tension glaucoma. In this group of patients, trabeculectomy will probably be a better option to slow the progression of glaucoma. The angle surgery as an option for other cases could enable better personalization of surgical glaucoma treatment. However, to place the GATT method right in the panel of antiglaucoma surgeries, longer observation time with an evaluation of its potential to stabilize glaucoma progression and more results of comparative studies are needed.

## 6. Conclusions

To sum up, a 6-month observation period showed that GATT is a minimally invasive glaucoma-surgery technique that makes it possible to achieve an effective and safe IOP decrease in patients with open-angle glaucoma.

## Figures and Tables

**Figure 1 ijerph-20-02759-f001:**
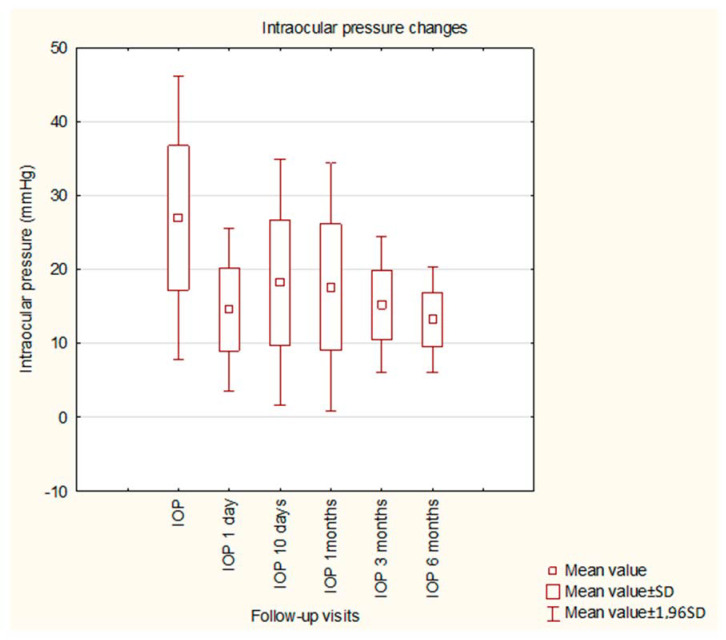
Intraocular pressure changes during the study.

**Figure 2 ijerph-20-02759-f002:**
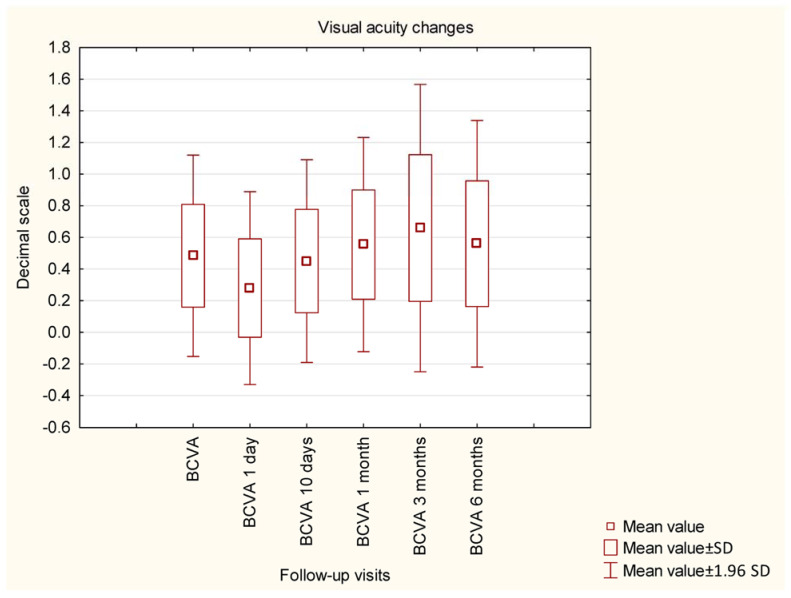
Changes in best corrected visual acuity during the study.

**Table 1 ijerph-20-02759-t001:** Demographic characteristics of the studied group.

Feature	
Number of patients	69
Age	69.48 +/− 11.30 years
Gender	35 F; 34 M
Type of glaucoma	48 POAG; 21 PEXG
IOP before surgery	26.94 mmHg +/− 9.70
VFI before surgery	58.36 % +/− 23.91
MD before surgery	−8.20 dB +/− 12.38
Number of drugs before surgery	2.59
BCVA before surgery (decimal scale)	0.47 +/− 0.32
Previous intraocular surgery	Cataract surgery: 27 Previous antiglaucoma surgery: 8 (trabeculectomy: 6; NPDS: 1; XEN: 1; MiniExpress: 1, revision of the bleb: 3)

**Table 2 ijerph-20-02759-t002:** Success criteria 1: 30% decrease in intraocular pressure after GATT.

30% Decrease in IOP	1D	10D	1M	3M	6M
**S**	79.7% (55)	66.7% (46)	62.3% (43)	59.4% (41)	57.9% (40)
**QS**	1.4% (1)	7.2% (5)	20.4% (14)	36.2% (25)	37.7% (26)
**F**	18.9% (13)	26.1% (18)	17.3% (12)	4.4% (3)	4.4% (3)

**Table 3 ijerph-20-02759-t003:** Success criteria 2: intraocular pressure lower than 18 mmHg after GATT procedure.

IOP ≤ 18 mmHg	1D	10D	1M	3M	6M
**S**	78.3% (54)	62.3% (43)	59.4% (41)	55.1% (38)	58.0% (40)
**QS**	1.4% (1)	4.3% (3)	18.8% (13)	36.2% (25)	33.3% (23)
**F**	20.3% (14)	33.4% (23)	21.8% (15)	8.7% (6)	8.7% (6)

**Table 4 ijerph-20-02759-t004:** Complications observed after GATT procedure.

Complication	Number/Percentage
Hyphema	35/69 (50.7%)
Transient visual loss	3/69 (4.3%)
Irydodialisis	2/69 (2.9%)
Corneal oedema	2/69 (2.9%)
Vitreous hemorrhage	1/69 (1.4%)
Additional antiglaucoma surgery	3/69 (4.3%)

## Data Availability

Data are available from corresponding author on request.

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
