# Peer review of "The Efficacy and Safety of the GATT Procedure in Open-Angle Glaucoma—6-Month Results"

_ijerph, 2023, doi:10.3390/ijerph20032759_

Round 1

Reviewer 1 Report

It is a very interesting topic and a well-written manuscript.

Title page: Please write the type of study in the title of your paper.

The introduction needs more information about various glaucoma treatments available and which patient needs surgical options for treatment. Please discuss GATT in more detail as per the current literature.

What were the inclusion and exclusion criteria for patient participation?  

You did not indicate which method of determining visual acuity you used (LogMAR or Snellen) and looks like the best corrected visual acuity you measured.

Need more references from the literature in the discussion section.

Author Response

Response to the Reviewer 1

It is a very interesting topic and a well-written manuscript.

The authors would like to thank the reviewer for all the comments and remarks. The changes in the main manuscript made according to the Reviewer 1 suggestion are marked with green bold.

Title page: Please write the type of study in the title of your paper.

The study was designed retrospectively, which was added in the aim paragraph of the abstract according to the Reviewer’s suggestion.

The introduction needs more information about various glaucoma treatments available and which patient needs surgical options for treatment. Please discuss GATT in more detail as per the current literature.

The following paragraph was added to Introduction section according to the Reviewer’s suggestion.

During the surgery the Schlemm’s canal could be achieved from the scleral approach (ab externo) or from anterior chamber (ab interno). The ab externo procedures need conjunctival and scleral opening and after reaching the Schlemm’s canal lumen, it can be dilatated by the injection of viscoelastic material and the suture tension (as in canaloplasty) or its internal wall may be removed (trabeculotomy 3600). The ab interno approach spares conjunctiva and sclera and the techniques are referred to as minimally invasive glaucoma surgery (MIGS) methods. There is also a set of ab interno methods with canal lumen viscodilatation (canaloplasty), implants injection ( iSTENT or Focus), local incision of internal wall (goniotomy with Trabex or Kahook Dual Blade knife) or extended removal of internal wall and they include gonioscopy-assisted transluminal trabeculotomy (GATT).

What were the inclusion and exclusion criteria for patient participation?

The following data regarding inclusion and exclusion criteria were included into the manuscript according to the Reviewer’s suggestion.

All patients with primary open angle glaucoma or pseudoexfoliative glaucoma who underwent GATT and had at least 6 mont observation were included. The exclusion criteria were as follows: neovascular glaucoma, traumatic glaucoma, narrow or closed angle in gonioscopy, normal tension glaucoma, secondary glaucoma different than PEXG, age under 18 years

You did not indicate which method of determining visual acuity you used (LogMAR or Snellen) and looks like the best corrected visual acuity you measured.

Patients’ visual acuity was assessed according to Snellen decimal scale. The information was added to the methods section according to the Reviewer’s suggestion.

Need more references from the literature in the discussion section.

The GATT is relatively new surgical method, when put in Pubmed combination of phrases GATT and open angle glaucoma, 59 articles up to 2022 are available, but when you exclude juvenile, silicon oil, neovascular glaucomas and non-English language publications only 21 is left. Almost all of them were cited in the Discussion section.

Reviewer 2 Report

The authors have conducted a well designed study on GATT. These procedures are well established and there is no contraversies on their efficacy so it would be better to compare them with other modalities to add on the value of the findings. 

-----------------

The authors have conducted a study on the efficacy of GATT in glaucoma patients, GATT is a well Stablished procedure in treatment of glaucoma, so if the authors had a control group would be more helpful to compare their findings with a standard treatment. Also 6 months of follow up seems too short to expect to see changes in visual fields and this need to be mentioned as another weakness of the study.

Author Response

Response to the Reviewer 2

The authors have conducted a well-designed study on GATT. These procedures are well established and there is no controversies on their efficacy so it would be better to compare them with other modalities to add on the value of the findings. 

The authors would like to thank the reviewer for all the comments and remarks. The changes in the main manuscript made according to the Reviewer 2 suggestion are marked with blue bold.

The authors have conducted a study on the efficacy of GATT in glaucoma patients, GATT is a well-established procedure in treatment of glaucoma, so if the authors had a control group would be more helpful to compare their findings with a standard treatment.

The first comparative studies regarding GATT have been published within the last few months. The authors also designed and participate in the ongoing studies comparing GATT to trabeculectomy and different angle surgery methods. Non-comparative character of the study was added as the limitation of the study according to the reviewer’s suggestion. The following sentence was added to the last paragraph of the Discussion.

However, to place the GATT method right in the panel of antiglaucoma surgeries the longer observations time with the evaluation of its potential to stabilize glaucoma progression, and more results of comparative studies are needed.

Also 6 months of follow up seems too short to expect to see changes in visual fields and this need to be mentioned as another weakness of the study.

To monitor the potential of the technique to stabilize glaucoma IOP and VF results are needed. The 6 month results focus mainly of the IOP observation. It is generally approved that you need to have 2 year results to detect VF progression. The following sentence was added to the Discussion paragraph according to the Reviewer’s suggestion.

However, to place the GATT method right in the panel of antiglaucoma surgeries the longer observations time with the evaluation of its potential to stabilize glaucoma progression, and more results of comparative studies are needed.
